# Optimization of Soybean Protein Extraction with Ammonium Hydroxide (NH_4_OH) Using Response Surface Methodology

**DOI:** 10.3390/foods12071515

**Published:** 2023-04-03

**Authors:** Ibrahim Bello, Adewale Adeniyi, Taofeek Mukaila, Ademola Hammed

**Affiliations:** 1Agriculture and Biosystems Engineering, North Dakota State University, Fargo, ND 58102, USA; 2Environmental and Conservation Science, North Dakota State University, Fargo, ND 58102, USA

**Keywords:** soybean proteins, optimization, protein hydrolysis, ammonium hydroxide, hydrolysis, amine concentration

## Abstract

Plants have been recognized as renewable and sustainable sources of proteins. However, plant protein extraction is challenged by the plant’s recalcitrant cell wall. The conventional extraction methods make use of non-reusable strong alkali chemicals in protein-denaturing extraction conditions. In this study, soy protein was extracted using NH_4_OH, a weak, recoverable, and reusable alkali. The extraction conditions were optimized using response surface methodology (RSM). A central composite design (CCD) with four independent variables: temperature (25, 40, 55, 70, and 85 °C); NH_4_OH concentration (0.5, 1, and 1.5%); extraction time (6, 12, 18, and 24 h) and solvent ratio (1:5, 1:10, 1:15 and 1:20 *w*/*v*) were used to study the response variables (protein yield and amine concentration). Amine concentration indicates the extent of protein hydrolysis. The RSM model equation for the independent and response variables was computed and used to create the contour plots. A predicted yield of 64.89% protein and 0.19 mM amine revealed a multiple R-squared value of 0.83 and 0.78, respectively. The optimum conditions to obtain the maximum protein yield (65.66%) with the least amine concentration (0.14 Mm) were obtained with 0.5% NH_4_OH concentration, 12 h extraction time, and a 1:10 (*w*/*v*) solvent ratio at 52.5 °C. The findings suggest that NH_4_OH is suitable to extract soybean protein with little or no impact on protein denaturation.

## 1. Introduction

With the recent global population growth, the challenge of hunger and malnutrition has become a major concern affecting over two billion people especially in the underdeveloped world [1]. Dietary protein is a major nutrient that aids the growth, development, and overall wellbeing of the body. The global protein demand is, thus, projected to quadruple by 2050 [2,3,4]. Proteins from animals and plants are the most common sources of dietary protein [1,5,6,7]; although, it is claimed that animal-based protein sources are more popular [8]. However, plant-based protein sources are gaining ground as, compared to animal protein sources, they are more renewable, sustainable, cheaper, environment friendly and healthier [8,9,9,10,11]. With the continuous rise in consumer awareness, the motivation for a healthier diet, and concerns for global greenhouse gas emissions [12], consideration and the transition to plant-based proteins is on the increase [1,13]. Plant-based alternative food proteins, especially legumes, have, thus, been identified as potential solutions to the protein security challenge [1].

Soybean is one of the oldest leguminous plant-based protein sources for human and animal nutrition [14]. It is a good nutrient source in animal diets, containing approximately 50% protein, 21% oil, 30–35% carbohydrate and 4% minerals [15]. Soybean protein is a highly nutritious and beneficial protein source that contains essential amino acids [16] and has the potential to enhance the immune system [17,18]. It is a high-quality protein with a 100% amino acid digestibility [19], which is comparable to proteins from animal sources, such as meat, dairy, and egg [20], making it an excellent choice among plant-based proteins [19]. Although soy proteins lack some sulfur-containing amino acids, such as methionine, they contain all other essential amino acids, including valine, threonine, leucine, lysine, phenylalanine, isoleucine, tryptophan, and histidine. Due to its good amino acid profile and high protein content [15], soybean protein is widely used in various applications, including food formulations, feed, health, and chemical applications. Soybean is also a primary source of high-value secondary co-products, such as lecithin, vitamins, nutraceuticals, antioxidants and is a good substitute for lactic proteins [21] hence it is utilized by lactic acid intolerant patients. Due to these multiple functional properties, soybean has become a preferred vegetable protein for food applications.

Plant protein production is not only dependent on a reliable source but also a sustainable and efficient extraction process. Several extraction technologies have been used to obtain soybean protein. The use of an enzymatic process has been reported to increase soybean protein yield [21,22,23]. However, enzymes are expensive, making enzyme-aided extraction industrially unfeasible [24]. The use of acidic conditions during extraction has been reported [25,26] but the protein yield is less due to the protein’s poor solubility and the resulting protein suffers degradation due to peptide bond hydrolysis. In addition, hydrochloric acid extraction leads to the formation of chlorohydrin compounds, such as 3-chloro-1, 2-propanediol (MCPD) and 1, 3-dichloro-2-propanol (DCP) [27,28], which are a group of contaminants labelled by the Food and Agriculture Organization (FAO) and the World Health Organization (WHO) as genotoxic animal carcinogens [29].

The use of alkali conditions for soy protein extraction have also been reported [30]. Alkali extraction, unlike acid extraction, aids in the unbounding of proteins, takes less time, is cost effective, and is less prone to the formation of toxins or inhibitory products [31,32,33]. Although NaOH creates an alkali condition that improves protein solubility and yield, NaOH is a strong alkali and also causes degradation [34]. Therefore, a mild alkaline application with less severity is needed. NH_4_OH is a weak base that creates a mild alkali processing condition, which could potentially reduce protein denaturation [35]. Additionally, NH_4_OH is volatile [36] and can be easily removed by evaporation. It can be recovered and reused through distillation, thus, saving processing costs and preventing the release of toxins into the environment [37]. The effectiveness of the extraction method is affected by various factors, such as solvent concentration, temperature, extraction time, solid-to-solvent ratio, and the particle size of the plant matrix [38]. The application of unsuitable conditions in the extraction process reduces the efficacy of extraction [39]. Therefore, the extraction conditions must be chosen as to improve the yield of the desired response. This can be achieved through the optimization of process parameters. Numerous parameters, such as the solvent ratio, pH, temperature, and time, significantly affect protein extraction [40]. When numerous factors and interactions have an impact on the intended response(s), the response surface methodology (RSM) is a useful approach for process optimization [41].

The response surface methodology (RSM) is a mathematical and statistical method used to plan and analyze experiments. The main goal of RSM is the optimization of chemical reactions to achieve the highest possible yield at a minimal cost, in the shortest time, and of the highest purity. Regression equations are developed to determine the relationship between the product’s yield (dependent variable) and the input variables (independent variables) with a view to producing a better yield [42]. In this study, RSM, using the central composite design (CCD), was used to optimize the process parameters (solvent concentration, temperature, time, and solid-to-solvent ratio) for protein extraction from soybean. Extraction enhances the availability and suitable utilization of proteins [1,43]. Apart from being utilized as stabilizers, emulsifiers, and foaming agents in the industry, proteins also serve as nutrient fortifiers to enhance the nutritional value of finished products [44]. 

Although the process of protein extraction leads to hydrolysis, which negatively impacts the protein’s nutritional quality [45], the extent at which the protein is hydrolyzed during the extraction process is not known. For soy protein to retain its favorable organoleptic and functional qualities, the degree of hydrolysis must be generally low [41]. Therefore, to derive extraction conditions that would cause the least degradation to the extracted protein, amine quantification was simultaneously carried out with the protein extraction optimization to determine the optimal point at which the protein is extracted with the least degradation. To the best of our knowledge, this is the first time NH_4_OH soy protein extraction has been performed while also monitoring the degree of hydrolysis (amine concentration) in order to minimize protein denaturation.

## 2. Materials and Methods

### 2.1. Plant Material

Dried soybean seeds were obtained from the North Dakota State University (NDSU) pilot plant. Prior to their use, the soybean seeds were cleaned and manually sorted. After sorting, the seeds were ground and sieved. Soybeans of sieve mesh size 0.425–1 mm were used in this study.

### 2.2. Proximate Composition

Proximate analysis gives an overview of the quantity of macromolecules present in a sample [46,47,48,49]. It is, therefore, an important step towards protein extraction. The quantity of protein, moisture, fat, ash, and carbohydrates were determined (Table 1).

#### 2.2.1. Moisture Content Determination

Moisture content was determined using the oven drying method [21]. The powdered soybean sample (1 g) was weighed in an empty pan and dried using a Binder ED-56 oven dryer at 105 °C overnight. After allowing it to cool in a desiccator, the weight of the dried sample was measured. The percentage moisture content of the sample was determined using Equation (1).
(1)$$\mathrm{Percentage}\;\mathrm{moisture}\;\mathrm{content}=\frac{M_{m}}{W_{s}}\times 100$$
where $M_{m}$ is the mass of the moisture (weight loss) and $W_{s}$ is the weight of the sample.

#### 2.2.2. Total Protein Content

Crude protein content was determined using a nitrogen combustion protein analyzer (LECO FP5820) according to the AACC method 46-30.01 (1999). About 5 g of sample was subjected to pyrolysis (to make the nitrogen available in its free state) followed by combustion at 850 °C in the presence of pure oxygen. Nitrogen given off (% nitrogen) was detected using a thermal detector. The crude protein was then determined from the % nitrogen using a standard conversion factor of 6.25, as shown in Equation (2) (AACC method 1999).
(2)Crude protein, %=%N×6.25
where *N* is the nitrogen content.

#### 2.2.3. Total Ash Content

Ashing was performed using the Thermo Fisher Scientific Thermolyne Benchtop^TM^ 1100 °C Muffle Furnace (Vernon Hills, IL 60061, USA). The sample was weighed into empty crucibles and placed in the muffle furnace at 550 °C for 24 h [50]. After allowing it to cool in a desiccator, the weight of the ash was measured. The percentage ash content of the sample was determined using Equation (3) [50].
(3)$$\%\mathrm{Ash}\;\mathrm{content}=\frac{M_{a}}{W_{s}}\times 100$$
where $M_{a}$ is the mass of the ash and $W_{s}$ is the weight of the sample.

#### 2.2.4. Total Fat

The proximate analysis of fat was performed using a Dionex ASE 200 Accelerated Solvent Extractor (Poway, CA, USA). About 6 g of sample was weighed into empty iron vials with cellulose filter papers inserted at their bases. The vials were tightly closed and transferred into the accelerated solvent extractor (using hexane as the extraction solvent) at 1000 psi [51]. The resulting oil–hexane mixture was aerated at 60 °C in a water bath to evaporate the remaining hexane. The oil was then transferred into the vacuum oven overnight to dry. The weight of the oil was recorded.

#### 2.2.5. Total Carbohydrate

Carbohydrate determination was performed using the percentage of carbohydrate by subtraction method. This is achieved based on the calculation of the difference between 100 and the sum of the percentages of moisture, ash, fat, and protein, as shown in Equation (4) [52].
(4) Percent of Carbohydrate=100%−sum of other proximate components

### 2.3. Experimental Design 

The effect of the four independent variables on protein yield and amine concentration was investigated using the central composite design (CCD) and response surface methodology (RSM). A total of thirty experimental runs for the optimization of the extraction parameters were carried out. Five levels (−2, −1, 0, +1, +2) were used for each independent variable. The four independent variables are extraction time (X_1_), temperature (X_2_), NH_4_OH concentration (X_3_), and solvent ratio (X_4_), while the protein yield (Y_1_) and amine concentration (Y_2_) are dependent variables.

The independent variables with their levels and codes are shown in Table 2.

### 2.4. Analytics

#### 2.4.1. Protein Extraction

The central composite design experiment was setup with 30 experimental runs of independent variables—NH_4_OH solvent system (1, 2.5, 5, 10 and 15%), sample-to-solvent ratios, extraction time, and temperature. A total of 5 g of powdered soybean sample was weighed into an Erlenmeyer flask (100 mL). Then, 50 mL of each solvent concentration was poured into each flask and transferred into a shaker at 55 °C and 130 rpm. Extractions were conducted with times of 6, 12, 18, and 24 h. Collected extracts were centrifuged at 1000 rpm for 5 min and the particulate and soluble fractions were separated via decantation.

The pH of the soluble fractions (supernatant) was adjusted to the protein isoelectric point pH (4–4.5) using dilute HCl, and left overnight to precipitate [21]. The resulting solution was centrifuged at 1000× *g* for 10 min. The supernatant was discarded, and the precipitate was washed with distilled water and oven dried at 60 °C for 12 h [53] using the Binder ED-56 oven dryer (Horsham, PA, USA). The amount of protein extract was determined using the Bradford assay protocol for protein estimation [54]. The protein standard curve was established using the Bovine Serum Albumin (BSA) assay [54]. The absorbance was obtained with the use of a Tecan Infinite M Nano, single-mode microplate reader (Seestrasse Männedorf, Switzerland) at 595 nm wavelength. The stages of protein extraction are shown in Figure 1. 

#### 2.4.2. Total Amine Estimation

Amines react with TNBS (2,4,6-Trinitrobenzene Sulfonic Acid) assay to form chromogenic derivatives that can be quantified spectrophotometrically at 335 nm wavelength. The total amount of amine in the sample was quantified using the TNBS assay method [55,56]. The glycine standard (20 μg/mL) was prepared by dissolving 0.2 g of glycine in 10 mL of distilled water followed by appropriate dilutions. The soybean sample was dissolved in a 0.1 M sodium bicarbonate reaction buffer (pH 8.5). In total, 0.25 mL of 0.01% (*w*/*v*) solution of TNBS was added to 0.5 mL of the sample solution and vortexed. The solution was incubated at 37 °C for 2 h. Then, 0.25 mL of 10% SDS and 0.125 mL of 1 N HCl were added to each sample. The absorbance of the resulting solution was taken at the 335 nm wavelength.

### 2.5. Statistical Analysis

The experimental data generated were subjected to multiple regression analysis using open-source statistical package (R) software. An empirical linear and second-order polynomial (pure quadratic) model was used to fit the data generated. Experimental design, data analysis, optimization, and contour plotting were also performed with R statistical software version R-4.2.3. The following model was proposed for the yield:(5)$$Y=b0+b1X_{1}+b2X_{2}+b3X_{3}+b4X_{4}+b11X_{1}{}^{2}+b22X_{2}{}^{2}+b33X_{3}{}^{2}+b44X_{4}{}^{2}+b12X_{1}X_{2}+b13X_{1}X_{3}\;+b14X_{1}X_{4}+b23X_{2}X_{3}+b24X_{2}X_{4}+b34X_{3}X_{4}\;$$
where *Y* is the response (protein yield and amine concentration); *b*0 is the value of the fixed response at the central point; *b*1, *b*2, *b*3 and *b*4 are the coefficients of the linear terms; *b*11, *b*22, *b*33 and *b*44 are the coefficients of the quadratic terms; and *b*12, *b*13, *b*14, *b*23, *b*24 and *b*34 are the coefficients of the cross products (interactive terms).

## 3. Results and Discussion

### 3.1. Proximate Analysis

The results of the proximate analysis (Table 1) shows that the soybean sample contains 10.78% moisture, 31.50% protein, 19.34% fat, 33.84% carbohydrate, and 4.53% ash. The sample was found to have high protein, oil, and carbohydrate content, with the latter being largely due to the presence of pigmented pericarp [15,57], which can be difficult to grind. The amount of oil (19.34%) found in our sample can be attributed to the high amount of protein, which has the capacity to capture and retain oil [58,59]. These findings are consistent with earlier studies, which reported soybean to contain 10.74% moisture, 17.5% protein, 19.98% fat, 44.26% carbohydrate and 4.29% ash [60,61,62,63]. These results indicate that the proximate composition of the soybean sample used in this study is comparable to those used in previous studies.

### 3.2. The Response Surface Optimization

#### 3.2.1. Experimental of the Response Surface Methodology

The experimental protein extraction yield (Y_1_) and amine concentration (Y_2_) obtained from 30 experimental runs of four independent variables are shown in Table 3. The results showed that a maximum experimental protein yield of 99.88% was reached at positions X_1_, X_2_, X_3_, X_4_ = (1, 1, −1, −1), which corresponds to an amine concentration (Y_2_) of 0.23 mM. The least amine concentration (0.15 mM) was reached at conditions X_1_, X_2_, X_3_, X_4_ = (−1, −1, −1, 1), corresponding to a protein extraction yield (Y_1_) of 56.14%.

The independent and dependent variables were then analyzed using the developed model to create a regression equation that could predict the response within the specified range.

#### 3.2.2. Regression Models for Response Variables 

##### Regression Models for Protein Extraction Yield (Y_1_)

The data shown in Table 3 were subjected to multiple regression analysis using the quadratic interaction coefficients for protein extraction yield (Y_1_). Table 4 shows the analysis of variance (ANOVA) of the independent variables for the extraction optimization of soybean protein.

The goodness-of-fit and lack-of-fit test results following the regression model are also presented. The regression model equation for protein extraction yield (Y_1_) is shown in Equation (6) as follows:(6)$$Y_{1}=73.540-5.149X_{1}+19.063X_{2}-2.978X_{3}-4.446X_{4}-6.005X_{1}^{2}-2.280X_{2}^{2}+8.878X_{3}^{2}-14.220X_{4}^{2}$$

The statistical analysis results revealed that only the quadratic and linear interaction coefficients were significant (*p* < 0.05), while the two-way interaction coefficients with a high *p*-Value of 0.512 were not significant (*p* > 0.05) and were removed from the model.

The ANOVA goodness-of-fit, provided by the coefficient of determination (R^2^), was determined. R^2^ measures the percentage of changes in the response variable that can be attributed to independent variables and their interactions. It also evaluates how well a statistical model fits a given set of data. The closer the R^2^ value is to 1, the better the model matches the data. The multiple and adjusted R^2^ values for the regression model for protein extraction (Y_1_) were 0.83 and 0.77, respectively, showing that the model was adequate. To further confirm the model’s adequacy, the lack-of-fit error test was carried out. The lack-of-fit error test quantifies inaccuracies due to any flaw(s) in a model [64]. The lack of fit is not significant if the error probability, *p*, of the lack-of-fit F-statistic is larger than the confidence interval. In contrast, if the F-statistic of the lack-of-fit error is greater than the associated error probability, the lack-of-fit test is said to be significant, and the regression model is inadequate to explain the data [65]. In the present study, as shown in Table 4, the lack of fit was found to be non-significant (F = 5.24; *p* = 0.4965 > 0.05), indicating that the regression model for protein extraction, Y_1_, was sufficient in explaining the experimental data.

##### Regression Model for Amine Concentration (Y_2_)

The data shown in Table 3 was subjected to the multiple regression analysis using the quadratic interaction coefficients for amine concentration (Y_2_). The regression model equation for amine concentration (Y_2_) is shown in Equation (7) as follows:(7)$${\;Y}_{2}=0.214-0.000245X_{1}+0.00569X_{2}+0.0403X_{3}-0.0237X_{4}+0.0113X_{1}^{2}-0.0236X_{2}^{2}\;-0.011X_{3}^{2}+0.0196X_{4}^{2}$$

Table 5 shows the analysis of variance (ANOVA) of the independent variables for the degree of hydrolysis of protein (amine concentration). Statistical analysis data showed that the quadratic and linear interaction coefficients were significant (*p* < 0.05) while the two-way interactions were not significant (*p* > 0.05).

To acquire a satisfactory fit for a model, R^2^ values should be >0.80 [66]. The multiple and adjusted R^2^ values for the regression model for amine concentration (Y_2_) were 0.78 and 0.58, respectively. These values are quite low. However, since the error probability of the lack-of-fit F-statistic is larger than the confidence interval (F = 1.00; *p* = 0.531 > 0.05), the lack of fit was not significant [65], and the model shows adequacy in explaining the variations in data.

### 3.3. Surface Plots for Protein Extraction Yield (Y_1_) and Amine Concentration (Y_2_)

In this study, the optimum conditions were selected using surface plots. To determine the optimum yield at any points, two of the four independent variables were fixed while varying the remaining two and predicting the response variables. 

As shown in Figure 2, interactions between the solvent ratio and extraction time caused an increase in protein extraction yield, resulting in a maximum protein yield of about 65% at a time and solvent ratio of 12 h and 1:10 *w*/*v,* respectively. However, a further increase in the time and solvent ratio after reaching the optimum caused a corresponding decrease in protein yield. An increase in temperature and NH_4_OH concentration caused a corresponding increase in protein yield, as shown in Figure 3 and Figure 4, respectively. Interaction between the temperature and time showed that the protein yield was highest when the temperature was around 80 °C and the extraction time was 12 h (Figure 3). In addition, variations in the NH_4_OH concentration and solvent ratio showed that the protein yield was highest when the NH_4_OH concentration was 1.0%. However, at such higher temperatures and concentrations, the nutritional quality of the proteins would have been compromised [35,67,68]. A further increase in extraction time does not have much effect on the protein extraction yield in the selected range of study.

The U-shaped amine concentration surface plots indicate amine concentration minimization, which is a measure of the degree of protein hydrolysis. Results revealed that the minimum amine concentration was attained at extraction time and solid-to-solvent ratio of 12 h and 1:10 respectively (Figure 5). This finding indicates that a longer extraction time and higher solvent ratio did not cause any further decrease in amine concentration. Moreover, the influence of extraction time and temperature on amine concentration showed that the minimum amine concentration was obtained at a temperature of 52.5 °C and an extraction time of 12 h (Figure 6). The minimum amine concentration was also obtained at a solid-to-solvent ratio and NH_4_OH concentration of 1:10 and 0.5%, respectively (Figure 7). This suggests that, at these points, the extracted protein was least denatured. Although it was observed that a higher amount of soybean protein was extracted with an increase in NH_4_OH concentration and temperature [68]; this is detrimental to the nutritional quality and organoleptic properties of protein. Therefore, considering the degree of hydrolysis, which must be generally low [41], the maximum protein extraction was reached at an extraction time of 12 h, a solid-to-solvent ratio of 1:10 *w*/*v*, a NH_4_OH concentration of 0.5%, and a temperature of 52.5 °C.

Similar optimization results for the extraction of proteins from plants have been reported. Ref. [69] extracted proteins from soybean flour using the response surface methodology. In the study, the effects of extraction time, temperature, pH and NaOH concentration were found to be significant with an optimum protein yield of 48.30% at 70 °C, pH 12.68, and a 44.7 min extraction time. In a similar study on protein extraction optimization from lentil using response surface methodology, ref. [70] came up with an optimum protein yield of 14.5 g/100 g flour at a temperature of 22 °C, a time of 1 h, and a solid-to-solvent ratio of 1:10 (g/mL).

In a related study, ref. [71] investigated protein extraction from *Chlorella vulgaris* sp. with an alkaline solubilization and acid precipitation technique using the response surface methodology. In the study, the effects of independent variables, such as the precipitation time and pH, were found to be significant and at an optimum at 39.86 min and 3.2, respectively, with an overall protein yield of 81.0%. The results of protein extraction from watermelon seeds with sodium hydroxide using the response surface methodology has also been reported. In the study, ref. [40] discovered the significant effects of temperature, liquid/solid ratio, time and solvent concentration on protein yield. They came to the conclusion that these conditions were the best for extraction: a NaOH concentration of 1.2%, mixing time of 15 min, a temperature of 40 °C, and a solvent/meal ratio of 70:1. Likewise, ref. [41] investigated the effects of pH, time, temperature, and the liquid/solid ratio, on protein extraction from peanuts (*Arachis hypogea* L.). They discovered substantial effects of these variables on protein yield and concluded that the maximum protein yield was obtained at a pH of 8.0, solvent ratio of 8:1, a time of 30 min, and a temperature of 50 °C. In this research, the optimum conditions for protein extraction varied slightly from the reported values because, unlike other protein extraction processes, the optimum values of the protein extraction yield were determined at the point when the degree of protein hydrolysis was at the minimum, signifying little or no protein denaturation.

### 3.4. Validation Studies

The experiment was repeated using the optimum conditions (extraction time of 12 h, solid-to-solvent ratio of 1:10 *w*/*v*, NH_4_OH concentration of 0.5%, and a temperature of 52.5 °C) derived from the above study. The experimental protein yield at the optimum level was 65.66%, while the computed protein yield using the RSM regression equation was 64.89%. This result validates the regression model.

## 4. Conclusions

The regression model for protein and amine concentration optimization exhibited a non-significant lack of fit and a goodness of fit (R^2^) of 83% and 78%, respectively. This result was validated in the experimental vs. predicted protein extraction yield and amine responses. Following 30 experimental runs of temperature, NH_4_OH concentration, extraction time, and solvent ratio, the experimental protein yield with the least degree of hydrolysis was 65.66%. The surface plots showed that the maximum protein yield with the least degree of denaturation was obtained by extracting soybean with a 0.5% NH_4_OH concentration, 12 h extraction time, and a 1:10 (*w*/*v*) solvent ratio at 52.5 °C. It was observed that, at optimum conditions, the experimental protein yield and amine concentration varied slightly with predicted values. The experimental protein yield (65.66%) was slightly higher than the predicted yield (64.89%), while the experimental amine concentration (0.14 Mm) was slightly lower than the predicted yield (0.19 Mm). The closeness of both data (experimental and predicted) shows that the predicted model is more suitable for the experimental data. It is obvious that ~40% of proteins are still captured in the soybean fiber as crystals. This could be a great resource in various industrial applications. The optimization results suggest that NH_4_OH is suitable to extract soybean protein without causing degradation.

## Figures and Tables

**Figure 1 foods-12-01515-f001:**
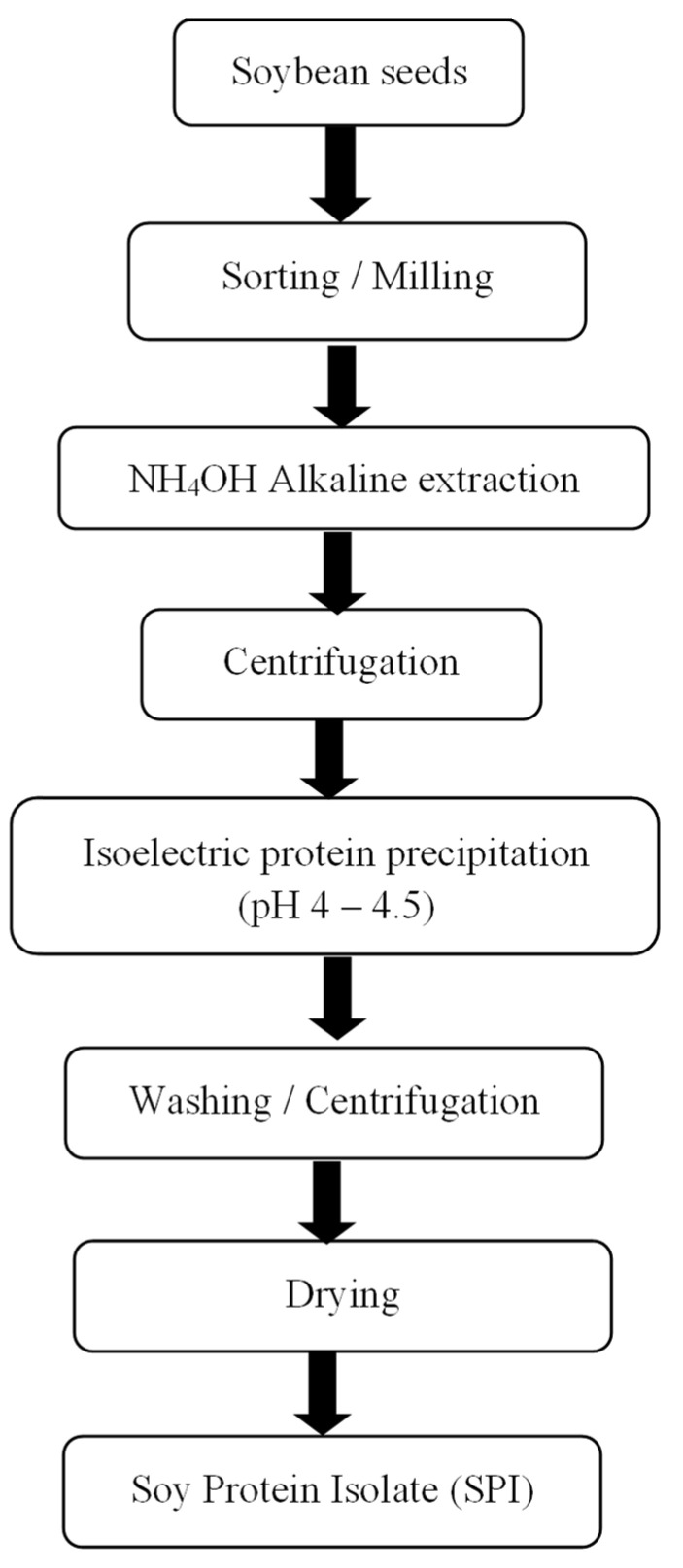
Simplified flowchart for the extraction of soybean protein.

**Figure 2 foods-12-01515-f002:**
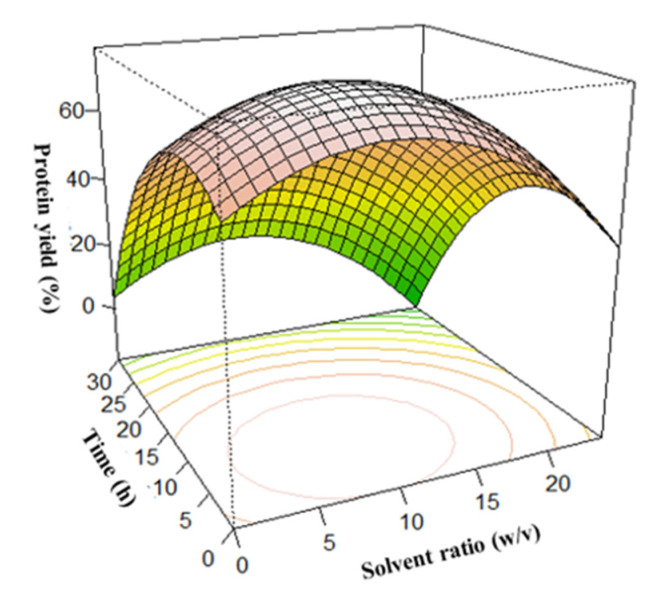
Effect of the extraction time and solvent ratio on protein yield.

**Figure 3 foods-12-01515-f003:**
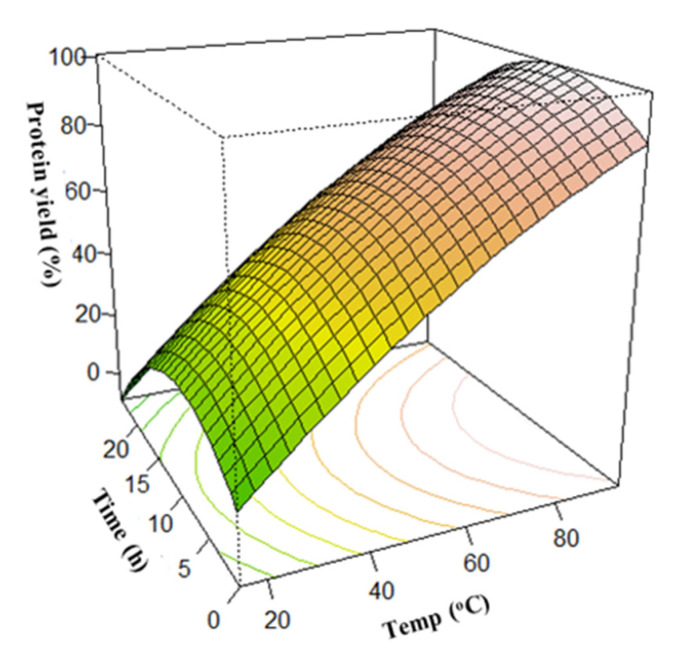
Effect of the extraction time and temperature on protein yield.

**Figure 4 foods-12-01515-f004:**
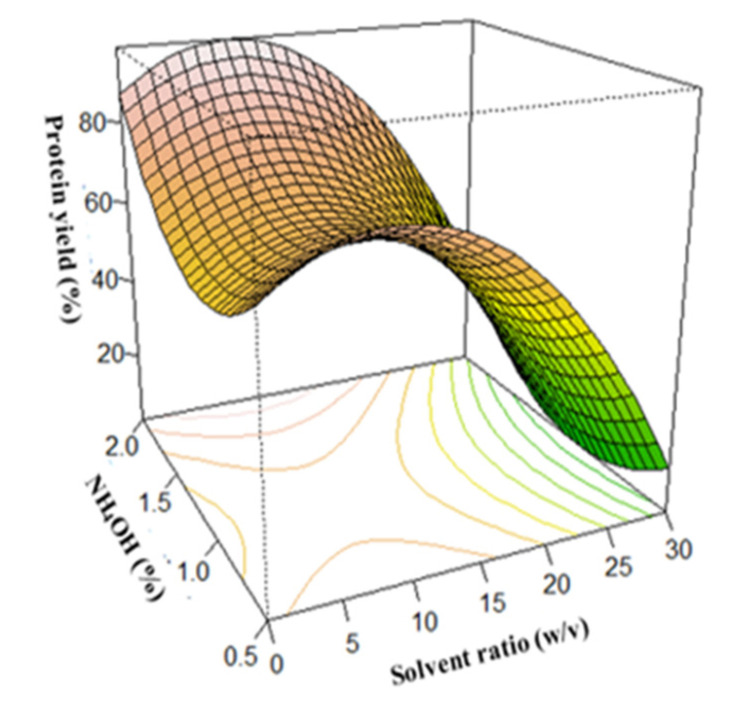
Effect of NH_4_OH concentration and solvent ratio on protein yield.

**Figure 5 foods-12-01515-f005:**
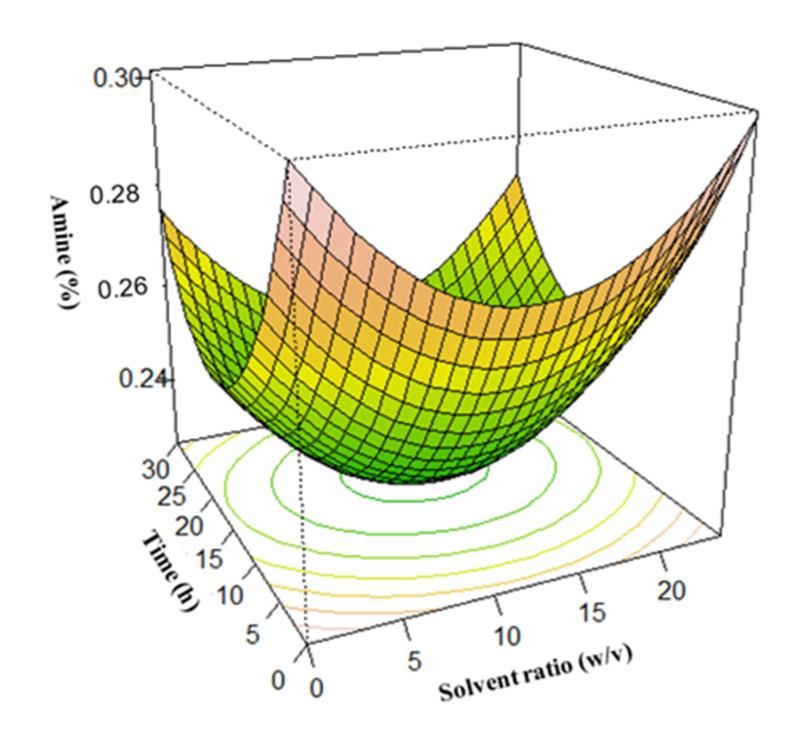
Effect of the extraction time and solvent ratio on amine concentration.

**Figure 6 foods-12-01515-f006:**
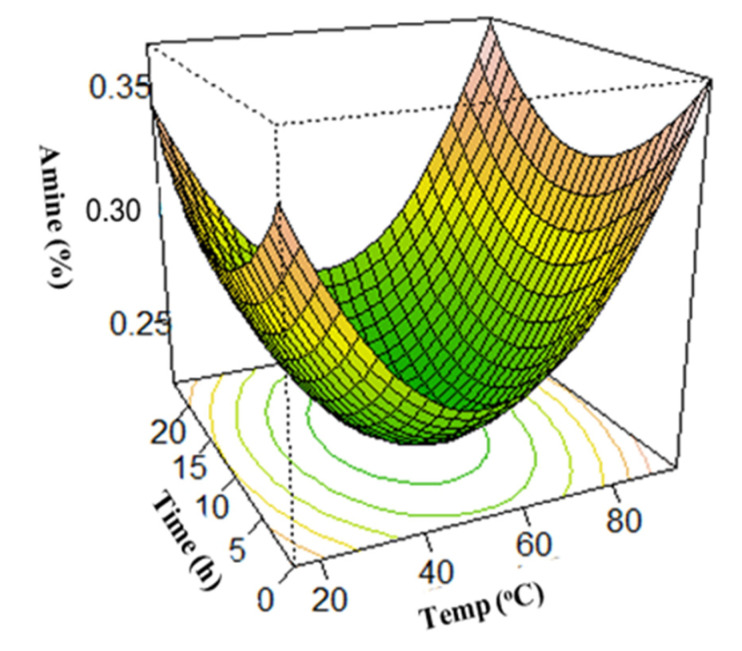
Effect of the extraction time and temperature on amine concentration.

**Figure 7 foods-12-01515-f007:**
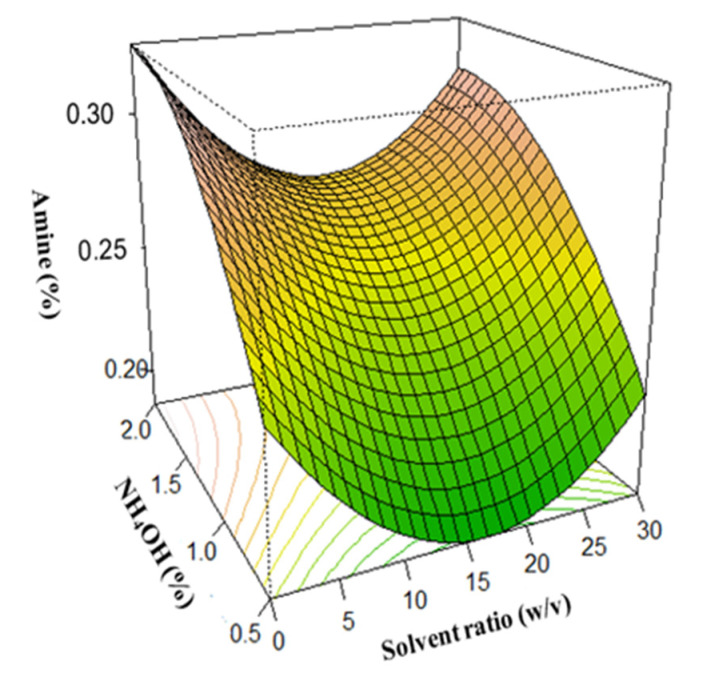
Effect of the NH_4_OH concentration and solvent ratio on amine concentration.

**Table 1 foods-12-01515-t001:** Proximate analysis of the soybean sample.

Parameters	Quantity (%)
Moisture	10.78 ± 0.07
Protein (*Ν* × 6.25)	31.50 ± 2.61
Fat	19.34 ± 0.11
Carbohydrate (by difference)	33.84 ± 1.67
Ash	4.53 ± 0.07

**Table 2 foods-12-01515-t002:** Independent variables and their levels in CCD.

Variables		Levels				
	Codes	−2	−1	0	1	2
Extraction time (h)	X_1_	0	6	12	18	24
Temperature (°C)	X_2_	25	40	55	70	85
NH_4_OH Concentration (%)	X_3_	0.5	0.5	1	1.5	2
Solvent ratio (*w*/*v*)	X_4_	1:5	1:5	1:10	1:15	1:15

**Table 3 foods-12-01515-t003:** Experimental design codes, actual values of the central composite design, and responses of the surface methodology for soybean protein extraction yield (Y_1_) and amine concentration (Y_2_).

	Coded Variables				Uncoded Variables				Responses	
Runs	X_1_	X_2_	X_3_	X_4_	Time, (X_1_) (h)	Temperature (X_2_), (°C)	NH_4_OH (X_3_), (%)	Solvent Ratio (X_4_)	Y_1_	Y_2_
									(%)	(mM)
1	−1	−1	−1	−1	6	40	0.5	5	44.51	0.25
2	1	−1	−1	−1	18	40	0.5	5	32.24	0.28
3	−1	1	−1	−1	6	70	0.5	5	84.10	0.23
4	1	1	−1	−1	18	70	0.5	5	99.88	0.23
5	−1	−1	1	−1	6	40	1.5	5	53.05	0.35
6	1	−1	1	−1	18	40	1.5	5	33.87	0.34
7	−1	1	1	−1	6	70	1.5	5	91.99	0.32
8	1	1	1	−1	18	70	1.5	5	79.09	0.28
9	−1	−1	−1	1	6	40	0.5	15	56.14	0.15
10	1	−1	−1	1	18	40	0.5	15	18.60	0.19
11	−1	1	−1	1	6	70	0.5	15	92.97	0.22
12	1	1	−1	1	18	70	0.5	15	78.56	0.21
13	−1	−1	1	1	6	40	1.5	15	41.99	0.27
14	1	−1	1	1	18	40	1.5	15	19.22	0.31
15	−1	1	1	1	6	70	1.5	15	94.22	0.30
16	1	1	1	1	18	70	1.5	15	53.65	0.28
17	0	0	0	0	12	55	1	10	70.46	0.23
18	0	0	0	0	12	55	1	10	75.50	0.17
19	0	0	0	0	12	55	1	10	64.89	0.19
20	0	0	0	0	12	55	1	10	83.04	0.22
21	−2	0	0	0	0	55	1	10	42.51	0.27
22	2	0	0	0	24	55	1	10	52.67	0.23
23	0	−2	0	0	12	25	1	10	41.82	0.24
24	0	2	0	0	12	85	1	10	83.16	0.35
25	0	0	2	0	12	55	2	10	99.23	0.23
26	0	0	0	2	12	55	1	20	3.90	0.23
27	0	0	0	0	12	55	1	10	73.67	0.24
28	0	0	0	0	12	55	1	10	73.31	0.22
29	0	0	0	0	12	55	1	10	80.78	0.18
30	0	0	0	0	12	55	1	10	66.67	0.27

**Table 4 foods-12-01515-t004:** Analysis of variance summary for the model interactions for protein yield, Y_1_.

Source	DF	Sum of Squares	Mean Squares	F-Value	*p*-Value
Regression Linear	14 4	16,736.4 11,063.7	1195.46 2765.93	7.672 18.2034	0.0001677 0.0000014
Two-way interaction	6	853.4	142.23	0.9127	0.512141
Quadratic	4	4819.3	1204.83	7.7317	0.0004647
Residuals	15	2337.5	155.83	-	-
Lack of fit	14	2912.7	208.05	5.2357	0.4965
Pure error	7	278.2	39.74	-	-
Total	35	19,927.3			

R^2^ = 83.27%; Adjusted R^2^ = 76.9%.

**Table 5 foods-12-01515-t005:** Analysis of variance summary for the model interactions for amine concentration, Y_2_.

Source	DF	Sum of Squares	Mean Squares	F-Value	*p*-Value
Regression Linear	14 4	0.0616806 0.0295495	0.0044 0.007387	3.855 6.4634	0.006919 0.003132
Two-way interaction	6	0.0075330	0.001255	1.0985	0.407582
Quadratic	4	0.0245981	0.006149	5.3804	0.006848
Residuals	15	0.0171443	0.001143	-	-
Lack of fit	14	0.0089026	0.001113	1.0000	0.531090
Pure error	7	0.0082416	0.001177	-	-
Total	35	0.0788224			

R^2^ = 78.25%; Adjusted R^2^ = 57.95%.

## Data Availability

Data is contained within the article.

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
