# Peer review of "Optimization of Soybean Protein Extraction with Ammonium Hydroxide (NH4OH) Using Response Surface Methodology"

_foods, 2023, doi:10.3390/foods12071515_

Round 1

Reviewer 1 Report

OPTIMIZATION OF SOYBEAN PROTEIN EXTRACTION WITH AMMONIUM HYDROXIDE (NH4OH) USING RESPONSE SURFACE METHODOLOGY

Authors: Ibrahim Adebayo Bello , Adewale Tola Adeniyi , Taofeek Mukaila , Ademola Hammed

The authors present a procedure based on response surface methodology for the optimization of Soybean protein extraction.

The paper contributes to a topic of increasing interest as it is process optimization through an experimental design procedure. In my opinion, the paper is worth to be published, although several items should be clarify.

1)    At 2.2 section, Proximate composition, it is stated that “The quantity of protein, moisture, fat, ash, and carbohydrates were determined (Table 2)”. But on Table 2, the moisture values are missing

2)    At 2.3 section, Experimental design:

a.    It is stated, “After the one-factor-at-a-time study previously conducted, the effect of four independent variables ……….”. Has this previous study been published?. If this is the case, the reference must be included. If the study is done as a previous step, information on how it was done should be included.

b.    At section 2.4.1. Protein extraction, it is stated that “The central composite design experiment was setup with 30 different combinations of independent variables”. As can be seen on Table 4, some of the runs are repetitions. For instance, run1 is done on the same combination of the four independent variables as in run2, and so on. So, although 30 runs are done, they correspond to 17 different combinations of independent variables, not to 30 since there are repetitions.

3)    At the results section,

a.     ANOVA tables (Table 5 and 6 should include all the information, that is, the Sum of Squares and Mean squares values.

b.    Additional information is required to properly interpret Table 5. Why/How the authors conclude that the two-way interactions coefficients were not significant?

c.     Why the authors use a significance level of a=0,01 instead of a=0,05 (usual value)?

d.    Why the two-way interactions were not considered for the regression model for amine concentration (Y2)?

Reviewer 2 Report

General overview.

Manuscript foods- 2266971 titled “Optimization of soybean protein extraction with ammonium hydroxide (NH4OH) using response surface methodology”, reported the parameter optimization (solvent concentration,  temperature, time, solid-to-solvent ratio, and particle size) for the extraction of soy protein using NH4OH. In detail the manuscript is very interesting however same additional explanation should be added.    

Major comment

Introduction

In different part of the text soy proteins are defined as high nutritional value. I suggest to add the aminoacidic profile highlighting the content of essential amino acids for human and animal nutrition.

Materials and methods

Add information about the selection of B samples considering the effect of particle size on protein extractions. It is not clear why the author selects the B sample. Add briefly explanation and references of previous study.

Table 2 check the moisture contents

Protein extraction: define NH4OH solvent system. In addition why the extraction was made only from B sample if in the introduction was reported that “In this study, RSM using the central  composite design (CCD) was used to optimize process parameters (solvent concentration, temperature, time, solid-to-solvent ratio, and particle size)”?

Result and discussion

the discussions of the results are too sparse. I suggest to add more.

The author could be explain haw the parameter affect the proteins extractions.  In addition what are the effects of extraction Time, Temperatures NH4OH (X3), (%) and Solvent ratio on protein structure/degradation?

Minor

Line 45 change It is a good protein source with It is a good nutrient source

Round 2

Reviewer 2 Report

no additional comment